behaviour/ecology/evolution

pair bonds, pair living, monogamy, titi monkeys, *Plecturocebus*

**Author for correspondence:**
Sofya Dolotovskaya
e-mail: s.dolotovskaya@gmail.com

# What makes a pair bond in a Neotropical primate: female and male contributions

Sofya Dolotovskaya[1,2], Sarah Walker[3]
and Eckhard W. Heymann[1]

[1]Behavioural Ecology and Sociobiology Unit, and [2]Primate Genetics Laboratory, German Primate Center, Göttingen, Germany
[3]Estación Biológica Quebrada Blanco, Quebrada Blanco, Río Tahuayo, Peru

SD, 0000-0002-1401-3764; EWH, 0000-0002-4259-8018

Pair living and pair bonding are rare in mammals, and the mechanisms of their maintenance remain a puzzle. Titi monkeys, a 'textbook example' for 'monogamous' primates, have strong pair bonds and extensive male care. To investigate mechanisms of pair-bond maintenance, we studied seven wild groups of red titis (*Plecturocebus cupreus*) in Peruvian Amazonia over a period of 14 months. We analysed pair bonds by measuring proximity, grooming and approaches/leaves within pairs, and collected data on intergroup encounters. Females contributed to grooming more than males, especially during infant dependency, when most of the grooming within pairs was done by females. Females were also more active in controlling proximity between pair mates, making most of the approaches and leaves. Males, on the other hand, invested more in territorial defences. They participated in more intergroup encounters than females and were more active during these encounters. Our data is most consistent with the 'male-services' hypothesis for pair-bond maintenance, where a female contributes more to the proximity and affiliation maintenance while a male provides beneficial services.

## 1. Introduction

Pair living, or social monogamy, is rare in mammals (3–9%: [1,2]) and still remains an evolutionary puzzle. In contrast to birds, where pair living and biparental care are very common (90%: [3]), gestation and lactation in mammals restricts offspring nourishment to females, resulting in a highly skewed parental investment. Males are thus expected to increase their reproductive success through mating with multiple females rather than increasing their parental investment and remaining with a single female [4,5].

Another mystery is why in some pair-living species adult males and females form pair bonds. Although the terms 'pair

bonding' and 'pair living' are often used interchangeably, here we will consider them as separate components of a social system [6,7]. We define pair living as a type of social organization where two opposite-sex adults share a home range or territory ('two-adult groups': [6,8,9]), and pair bonding as a type of social structure where adult male and female form a long-term (i.e. extending beyond one breeding season) affiliative relationship [10,11]. Pair living does not imply pair bonding but is often associated with it.

Pair bonds can be difficult to quantify for two reasons. First, pair-bond strength varies a lot between pair-living species [12]. Some species form 'dispersed' pairs: a male and a female share a common territory, but do not often interact and forage or sleep independently (e.g. maned wolves, *Chrysocyon brachyurus* [13]; red-tailed sportive lemurs, *Lepilemur ruficaudatus* [14]; fork-marked lemurs, *Phaner furcifer* [15]). In other species, a male and a female are almost permanently associated (e.g. Kirk's dik-dik, *Madoqua kirkii* [16]; Azara's night monkey, *Aotus azarae* [17]). Second, in species forming two-adult groups, in contrast to multimale–multifemale groups with identifiable heterosexual dyads, pair bonding can be confounded with pair living [11]. To quantify pair bonds and tease it apart from pair living, a set of 'diagnostic criteria' has been proposed: spatial relationship between pair mates, partner-specific behaviours and signs of distress during separation from the pair mate [18]. These behaviours, in turn, can be assessed by rates of affiliative interaction, proximity scores and measures of reciprocity between pair mates [19].

Yet another difficulty with quantifying pair bonds is that the exact set of behaviours included in the concept of pair bond can depend on the definition used. In a narrow sense, often used in the zoological literature, the pair bond is assessed by rates and the degree of symmetry of proximity and affiliation between pair mates [10,11,19]. When used in a broader sense, pair bond can also include territorial behaviours such as mate guarding or assistance in resource defence or infant care [11]. It is not easy to disentangle different functions of territorial behaviours, some of which (e.g. mate guarding) might be related to the pair-bond maintenance while others (e.g. interest in extra-pair mates) might not. Moreover, neither territorial behaviours nor allomaternal care imply the existence of pair bonds, since both can be present in species without pair bonding or pair living (e.g. mate guarding in red-tailed sportive lemurs living in 'dispersed' pairs [14] or male care in group-living Barbary macaques (*Macaca sylvanus* [20])). However, in the literature on pair bonds it is quite common to include all these behaviours in the set of pair-bond maintenance behaviours, especially when they occur in already established pairs (e.g. [20,21]).

There are many hypotheses to explain the evolution and maintenance of pair living and/or pair bonding in mammals (see, e.g. [2,7,11,22,23]). Here we discuss these hypotheses with regard to the pair bonding and focus on the explanations they suggest for its maintenance, rather than the evolutionary origins. We differentiate these hypotheses according to whether pair bonding provides benefits to only one or to both sexes and outline predictions they make regarding the female and male contributions to the pair bond.

(1) The 'resource-defence hypothesis': both a male and a female benefit from pair bonding to defend resources together [24]. Under this hypothesis, a male and a female are expected to be equally interested both in maintaining proximity and affiliation with a pair mate and defending their territory.

(2) The 'mate-defence hypothesis': bonding with a female is beneficial for a male when either the spatial distribution of females or the temporal distribution of fertile periods make it difficult for the males to defend access to more than one female at a time [25]. This hypothesis suggests that a male should be more interested in maintaining proximity and affiliation with a pair mate. Both sexes can contribute to the territorial defence, but for different reasons: while a male is expected to defend exclusive access to a female, a female can defend resources.

(3) The 'male-services hypothesis': a female benefits from bonding with a male when the male provides some important services such as territorial or antipredator defence, infant care or protection from infanticide by competing males [22,24,26]. Under this scenario, a female is expected to be more interested in maintaining proximity and affiliation with a pair mate while a male is expected to provide some significant services. This hypothesis does not make any assumptions about the territorial defence: while a male can invest in territorial defence to protect resources or infants, a female can participate in territorial defence as a form of mate guarding or to protect resources.

Neotropical titi monkeys (previously *Callicebus*; split into *Callicebus*, *Plecturocebus* and *Cheracebus*: [27]) are an excellent model to study the mechanisms of pair-bond maintenance. A textbook example of a

'monogamous' primate, titis form long-term pair bonds (at least up to 12 years, as shown in 12-year study of wild population of *Plecturocebus discolor* (previously *Callicebus discolor*), the longest dataset available so far; [28–31]). Titis live in groups comprising one reproductive pair and one to three offspring [29–33]. Pair bonds between adult males and females exhibit all 'diagnostic' characteristics mentioned above: spatial cohesiveness between pair mates, partner-specific behaviours (male–female duets), signs of distress during separation and strong preference for pair mates over strangers of either sex [34–36]. Adult male titis contribute heavily to infant care: the infant is carried almost exclusively by the social father and is returned to the mother only to suckle; males also play with offspring and share food with them more often than females [30,37–39].

The goal of our study was to examine the mechanisms of pair-bond maintenance in titis. Specifically, we wanted to assess (i) which factors affect rates of proximity and affiliation between pair mates, (ii) which sex contributes more to the proximity and affiliation maintenance, and (iii) which sex contributes more to the territorial defence. We examined grooming and proximity patterns within pairs and collected data on male and female participation in territorial defence in seven wild groups of red titis (*Plecturocebus cupreus*) in the Peruvian Amazon. We compare our results to the data from other pair-bonded mammals and discuss our findings in the broader context of evolution and maintenance of pair-bonding in mammals.

# 2. Methods

## 2.1. Study site and animals

The study was conducted at the Estación Biológica Quebrada Blanco (EBQB) in the north-eastern Peruvian Amazon (4°21′ S, 73°09′ W). We studied seven habituated titi groups in June–December 2017 and 2018. Group 1 had been habituated since 1997; the other groups were habituated during this study. On average, it took six (3–10) weeks to habituate a group. We began data collection only after the animals were fully habituated. We individually identified all the study animals based on the combination of body size, tail shape and coloration, and genital size and shape. During the study period, infants were born in five groups. Birth-dates and the composition of study groups is provided in electronic supplementary material, table S1. We defined infant dependency as the period until an infant was not carried by a male during group travel (at the age of *ca* 4.5 months: [37,40]). This also encompasses the period of most active lactation, as weaning begins when the infants are *ca* 4.5 months old [40].

## 2.2. Data collection

We followed each group in blocks of 5–6 days with the help of trained field assistants. In between periods of data collection, we monitored each group for 1–2 days a month for possible changes in group membership. We followed titis from the early morning when the animals left a sleeping site (or from when we located the group) until the late afternoon when the animals retired to a sleeping site (or until we lost them).

We used continuous focal animal sampling for the adult male and female of each group. We separated the focal samples on any given animal either by a focal sample of another animal or by, at least, a 10 min period. As focal animals were visible for variable periods of time, sampling periods varied from 3 min to 2 h. If the focal animal was out of view for more than 2 min, we terminated the observation. We discarded any samples where the focal animal was visible for less than 50% of time. We recorded social interactions (resting in body contact, active/passive grooming, and duetting; based on ethograms provided by [30]), the distance, and events of approaches and leaves (within 1 m) between pair mates.

We also recorded intergroup encounters scored when individuals of the study groups had visual contact with another group and responded to its presence by calling and/or chasing (in the wild, titis very rarely engage in direct physical attacks or fighting during the encounters, even though this has been occasionally observed in captivity [28,37,41,42]). We considered two encounters to be independent when all participants stopped calling and chasing for more than 30 min. We recorded the time, location, identities of participating groups and individuals, and the activities of participants (calling, chasing). Participation was scored when an individual was either calling, chasing, or both. If

in the beginning of an intergroup encounter an individual called alone and/or moved alone towards another group, we scored this individual as the initiator of the encounter.

## 2.3. Statistical analyses

To characterize rates of proximity and affiliation between pair mates, we calculated daily proportions of time pair mates spent in close proximity ($\leq 1$ m), including time spent in affiliative interactions (resting in body contact and grooming) for each pair.

### 2.3.1. Factors affecting rates of affiliation and proximity between pair mates

To examine which factors affect rates of proximity and affiliation between pair mates, we used a generalized linear mixed model (GLMM) [43] with beta error structure and logit link function. We used the presence of dependent infant (hereafter 'infant presence'), group size and season as fixed effects and group ID as a random effect. As a measure of seasonality, we used rainfall data (monthly averages in mm at Tamshiyacu (4°00′10.7″ S, 73°09′38.2″ W), *ca* 40 km from EBQB, available at https://www.worldweatheronline.com). We compressed the response to avoid zeros and ones using $y' = (y^*(n-1) + 0.5)/n$, where $n$ is the sample size [44]. To achieve an approximately symmetrical distribution, we further square root-transformed the response. We $z$-transformed group size and rainfall [45]. To reduce the probability of Type I Error [46] we included the random slope of rainfall within group and its correlation with the intercept. We tested the overall effect of infant presence, group size and season using a full-null model comparison based on a likelihood ratio test [47,48]. The null model lacked the fixed effects but was otherwise identical to the full model. We tested the fixed effects using likelihood ratio tests comparing the full model with reduced models excluding each of the effects one at a time [46]. To assess model stability, we compared the full model estimates with those obtained from models with the levels of the random effects excluded one at a time. The sample had 269 daily proportion values, taken from seven pairs.

### 2.3.2. Grooming reciprocity

To assess grooming reciprocity between pair mates, we first calculated the grooming reciprocity index for each pair [49]:

$$\frac{\mathrm{Gfm} - \mathrm{Gmf}}{\mathrm{Gfm} + \mathrm{Gmf}},$$

where Gfm is the amount of time that the female groomed the male and Gmf is the amount of time that the male groomed the female. The index ranges from $-1$ to 1; values closer to 1 indicate that a female grooms a male more than vice versa.

To further examine if grooming reciprocity is affected by infant presence, we used a GLMM with daily proportion of time a female groomed a male of the total grooming time between pair mates as a response. We compressed the response using the formula provided above [44]. The model design, including predictor transformations, was identical to that of the model described above, except for the correlation between the random slope and the random intercept being unidentifiable (as indicated by absolute values of 1) and thus excluded from the model. The null model used for full-null model comparison lacked the effect of infant presence. The sample had 103 daily proportion values, taken from seven pairs.

Both models were fitted in R (v. 3.5.3; [50]) using the package glmmTMB (v. 0.2.3; [51]). To check for collinearity between predictors, we determined variance inflation factors [52] with the function vif of the package car (v. 3.0.2; [53]). To assess model stability, we used a function kindly provided by Roger Mundry.

### 2.3.3. Proximity maintenance

To assess which individual was more responsible for maintaining proximity between pair mates, we first calculated the Hinde index [54]:

$$100 \times \left( \frac{\mathrm{Af}}{\mathrm{Af} + \mathrm{Am}} - \frac{\mathrm{Lf}}{\mathrm{Lf} + \mathrm{Lm}} \right),$$

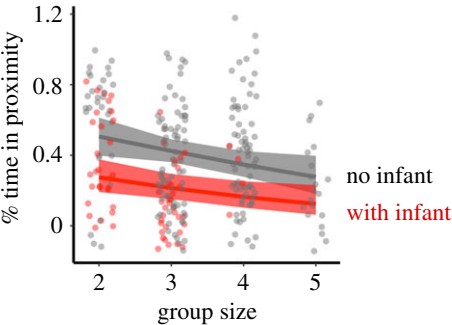

**Figure 1.** Daily proportion of time pair mates spent in close proximity as a function of group size, separately for the absence and presence of infant. The lines depict the fitted model (based on rainfall at its average), and grey and red areas show corresponding 95% confidence intervals.

where Af is the number of female approaches, Am is the number of male approaches, Lf is the number of female leaves and Lm is the number of male leaves. The index ranges from −100 to +100; high values indicate that proximity is mainly maintained by a female.

However, values of the Hinde index are difficult to interpret, since they do not indicate which individual makes most approaches and leaves, and different proximity patterns can thus result in the same values (a value of 0 can occur because female made equal number of approaches and leaves or because male made all approaches and leaves). To assess which individual is more active in maintaining proximity (makes more approaches and leaves), we calculated the Brown's index [55] using the same arguments:

$$100 \times \frac{Af + Lf}{Af + Am + Lf + Lm}.$$

The index ranges from 0 to 100; high values indicate that a female makes most approaches and leaves.

# 3. Results

## 3.1. Factors affecting rates of proximity and affiliation between pair mates

Infant presence and group size had a clear impact on rates of proximity and affiliation between pair mates (full-null model comparison $\chi^2 = 18.348$, d.f. = 3, $p < 0.001$). Specifically, pair mates spent less time in close proximity after infant birth ($\chi^2 = 16.524$, d.f. = 1, $p < 0.001$), and in larger groups pair mates spent less time in proximity than in smaller groups, although this effect was borderline significant ($\chi^2 = 3.759$, d.f. = 1, $p = 0.053$). Rainfall had no significant effect ($\chi^2 = 0.266$, d.f. = 1, $p = 0.610$) (electronic supplementary material, table S2; figure 1).

## 3.2. Grooming reciprocity

Overall, females groomed males more than vice versa, as indicated by the values of the grooming index closer to 1 (table 1). Grooming reciprocity between pair mates was further affected by infant presence (figure 2). While grooming was almost reciprocal before infant birth, females groomed males more than vice versa after infant birth (GLMM; likelihood ratio test comparing full and null model: $\chi^2 = 15.403$, d.f. = 1, $p < 0.001$).

## 3.3. Proximity maintenance

Females were more active in maintaining proximity, making the majority of both approaches and leaves within pairs as indicated by Brown's index and proportion of female approaches (table 1). The values of Hinde index provided mixed results, indicating females to be more responsible for maintaining proximity in some pairs and males to be more responsible in other pairs; overall, however, the values were not substantially different from 0 (on a scale from −100 to +100). To exclude the possibility that primarily female activity in maintaining proximity was caused by a lactating female addressing an infant carried by a male and not the male itself, we further calculated Brown's index separately for

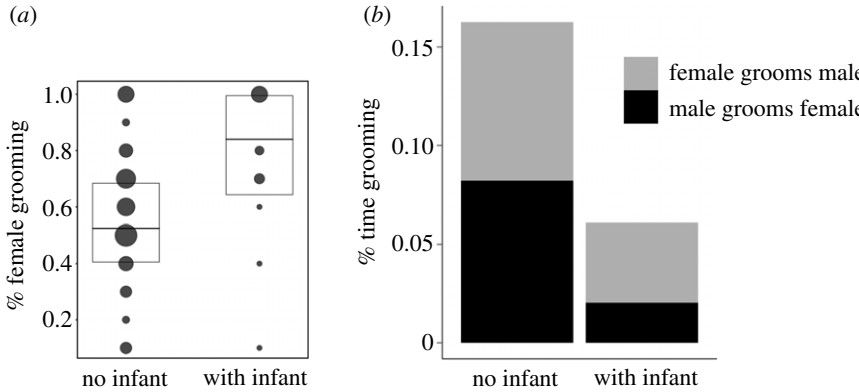

**Figure 2.** (*a*) Proportions of female investment in grooming within pairs, before versus after infant birth. For visual clarity proportion was binned into 10 sections. The area of the dots corresponds to the respective number of observations (0–18 per bin). Boxes depict median and lower and upper quartiles. (*b*) Mean daily proportion of time spent grooming within pairs, before versus after infant birth.

**Table 1.** Grooming, Hinde and Brown's indexes and proportion of female approaches from the total number of approaches within pairs.

| group | grooming index | Hinde index | Brown's index[a] | proportion of female approaches |
|---|---|---|---|---|
| 1 | 0.92 | −16.78 | 62.59 (76.90/59.29) | 0.56 |
| 2 | 0.96 | −20.84 | 60.91 (50.85/72.55) | 0.54 |
| 3 | 0.71 | −30.00 | 63.64 (63.64/[b]) | 0.50 |
| 4 | 0.74 | 14.10 | 68.46[c] | 0.74 |
| 5 | 0.99 | 8.97 | 65.91[c] | 0.69 |
| 6 | 0.99 | 6.67 | 64.29 (100.00/58.33) | 0.67 |
| 7 | 0.76 | −4.17 | 64.71 (64.29/66.67) | 0.63 |
| **Mean** | **0.87** | **−6.01** | **64.36** | **0.62** |

[a]Overall index with separate values for the periods with and without dependent infants, respectively, in parenthesis.
[b]Insufficient data to calculate the index.
[c]Groups only observed in the absence of dependent infants.

periods with and without dependent infants; values for both periods were still higher than 50.0, indicating that females were more active in the relationships regardless of the infant presence.

## 3.4. Intergroup encounters

Of 21 observed intergroup encounters, nine were initiated by a male and 12 did not have a clear initiator (for the full account of the encounters see electronic supplementary material, table S3). We never observed a female initiating an encounter. Males participated in all encounters, while females participated in 19 encounters. Males were more active during the encounters: in all 16 encounters for which the chasing data could be collected, males were both calling and chasing. In contrast, females mainly just called (16 encounters) and only chased during two encounters. We never observed a female chasing unless her mate was chasing, too.

## 4. Discussion

Overall, rates of proximity and affiliation between pair mates in red titi monkeys were affected by the presence of dependent infants and group size. After infant birth, pair mates spent less time in close proximity. A similar effect of infant presence was also demonstrated in a field study on *P. discolor* [39]. The decrease in time pair mates spend in proximity after infant birth is probably related to energetic costs of infant care that are high both for lactating females and carrying males [56] or to the fact that

males, while spending substantial amount of time socializing with infants [30,37–39], have less time available for their pair mates. Similarly, in larger family groups, i.e. those including juvenile and subadult offspring, with more potential social partners pair mates spent slightly less time in proximity than in smaller groups.

Females and males contributed differently to the pair-bond maintenance. Females contributed more than males to proximity and affiliation maintenance. First, they groomed males more than vice versa, especially during the period of infant dependency. Second, females were more active in controlling proximity, making the majority of approaches and leaves within pairs as indicated by Brown's index and proportion of female approaches, suggesting more female initiative and 'interest' in proximity. Males, on the other hand, contributed more than females to the territorial defence: they participated in more intergroup encounters and were more active during these encounters.

Primarily female contribution to the proximity and affiliation maintenance was demonstrated in most of the field studies on other titi species: in *Plecturocebus toppini* (previously *Callicebus brunneus*) [57], *Cheracebus torquatus* [38] and *P. discolor* [58] females groomed males more than the reverse (although in one study on *P. discolor* grooming was reciprocal [39]). In captive *Plecturocebus cupreus* (previously *Callicebus moloch*), females were more attached to males than vice versa: they spent more time than males close to experimental partitions physically separating pair mates [34], preferred a pair mate to an empty cage or a stranger male more often than males did [59], and were more reluctant than males to interact with opposite-sex strangers [21]. Approach rates and the Hinde index provided mixed results in titis: while in our study females approached males more often in all pairs, it was true only for some pairs in *P. toppini* [57], and males approached more often in *P. discolor* [29]. The Hinde index indicated neither sex to be more responsible for maintaining proximity in our study and in *P. toppini* [57], but showed males to be more responsible in *P. discolor* (although the bias was not very pronounced, as indicated by low index values: 18 and 25, respectively [28,30]). Unfortunately, none of these studies calculated the Brown index, and the difficulty of interpreting the values of Hinde index (see Methods) does not allow to assess which sex was more active in the relationship.

Primarily male contribution to the territorial defence is consistent with other titi studies. In *P. discolor* and *P. toppini*, males participated in more intergroup encounters than females, initiated them more often, and called and chased more during the encounters [37,41,57,60]. In *P. discolor*, males initiated the duetting near group boundaries more often than females, and responded stronger (i.e. initiated response duetting more often) than females to the simulated duets in playback experiments, indicating more active male involvement in the boundary reinforcement [60]. In captive *P. cupreus*, males show more agitation and distress than females in the presence of intruders of both sexes [21,36,59,61].

Our observations, together with data on other titi species, are most consistent with the 'male-services' hypothesis that predicts that a female would show more initiative and 'interest' in maintaining proximity and affiliation with a pair mate in exchange for some important services provided by a male. This hypothesis is further supported by a fact that grooming between partners was more heavily skewed towards female investment during the period of infant dependency, when male services are most needed. While males reduced the amount of grooming directed at females after infant birth, females conserved the amount of time they groomed males, suggesting the importance of maintaining proximity and affiliation with pair mates for the females. So which services does a male provide?

First, male titis provide extensive infant care, releasing the lactating females of all the costs of infant carrying, sharing food and socializing with them. Second, they provide anti-predator defence: both in our study groups [62] and in *P. discolor*, males were more active during encounters with predators. Although sex differences in vigilance have not been quantified for titis yet, both in our study (2018, unpublished data) and in *P. toppini* [57] males appeared to be more vigilant than females. By providing anti-predator defence, a male allows a female to focus on foraging [24]. Interestingly, in *P. discolor* males demonstrated active anti-predator behaviours only in the presence of infants [63].

Finally, males provide territorial defence. The function of this behaviour is probably mixed and can represent resource defence, mate defence, or both. Playback studies trying to tease apart these two functions of territorial defence provided somewhat more support for resource defence. In *P. toppini*, males reacted stronger to playbacks in the high-used versus low-used parts of the home range [57]. In *P. discolor* [60] and *Callicebus nigrifrons* [63], males did not react stronger to playbacks of male solos than to playbacks of duets. Finally, in *C. nigrifrons*, pairs were not duetting more often during the periods of likely female fertility [64]. Mate defence received only weak support in *P. toppini*: males reacted stronger when duets were played closer to their mates [57]. In captive *P. cupreus*, however, males clearly demonstrated mate-guarding behaviour: they showed increased attraction to a pair mate

**Table 2.** Intensity of male care and sex investment in the proximity and affiliation maintenance for pair-bonded mammals based on data from field studies. Male care: N, no care; M, moderate care; I, intense care (following classification criteria in [17]).

| species | male care | which sex contributes more to proximity and affiliation maintenance | measures of contributions used | references |
|---|---|---|---|---|
| *Madoqua kirkii* (Kirk's dik-dik) | N | males | approach/leave data | [16] |
| *Hylobates lar* (white-handed gibbon) | N | males | grooming reciprocity, approach/leave data | [67] |
| *Indri indri* | N | males | grooming reciprocity | [68] |
| *Pithecia pithecia* (white-faced saki monkey) | N | females | grooming reciprocity, approach/leave data | [29,69] |
| *Symphalangus syndactylus* (siamang) | M | both sexes | grooming reciprocity, approach/leave data | [67] |
| *Petropseudes dahli* (rock-haunting possum) | I | both sexes | approach data | [70] |
| *Otocyon megalotis* (bat-eared fox) | I | both sexes | approach data | [71,72] |
| *Aotus nancymaae* (owl monkey) | I | both sexes[a] | grooming reciprocity, approach/leave data | [73,74] |
| *Plecturocebus cupreus* (red titi monkey) | I | females | grooming reciprocity, approach/leave data | this study |

[a]Data available only for captive animals.

and agonism towards a male intruder as a function of increasing proximity between the pair mate and the intruder [59,61].

It is likely that participation in the intergroup encounters serves both for resource and mate defence, as these functions are not mutually exclusive. Territorial defence ensures exclusive use of space, which in turn allows exclusive access to both resources and mates [65]. In this respect, it should be noted that females participated in most (19 of 21) intergroup encounters together with males, even if they were not as active as males and, unlike males, almost never chased the animals from the neighbouring group. Female participation in encounters provides some support for the 'resource-defence' hypothesis where both sexes defend their territory together. However, more active male participation in territorial defence together with more pronounced female contribution to the proximity and affiliation maintenance provide arguments in favour of the 'male-services' hypothesis.

Another likely reason for the males to participate in the intergroup encounters represents the other side of the mate defence: an interest in extra-pair mates. This possibility cannot be ruled out either for males or females. There is one report on extra-pair copulations in titis [28] and several reports on mate displacements [57,66]. In the field [57,60] and captive [34,59] studies, both sexes demonstrated mate-guarding behaviour (e.g. responded stronger to the same-sex playback calls than to opposite-sex calls), although males to a greater extent. Pair mates were also more affiliative during the intergroup encounters in *P. toppini* and *P. discolor* [57,60], a behaviour probably enabling both sexes to guard their partners from potential extra-pair mates.

Comparison with other pair-bonded mammals suggests an association between the intensity of male care for infants and the pattern on pair-bond maintenance (table 2). Generally, the more intense male care is, the more a female contributes to the maintenance of proximity and affiliation with a male. While in species with no male care males are primarily responsible for proximity and affiliation maintenance,

in species with moderate or intense male care females contribute to proximity and affiliation maintenance equally or more than males. The only exception is sakis, where females contribute to proximity and affiliation maintenance more than males despite the complete absence of male care. However, it has been shown that male sakis contribute more than females to territorial and anti-predator defence, especially during the infant dependency [63,69], possibly providing indirect benefits to females. Interestingly, like in our study, the skew towards female contribution to proximity and affiliation maintenance was more pronounced during the period of infant dependency [69]. This might indicate a female's increased value of male services during the period when these services are most needed. Male care has been suggested as a driver for the evolution of pair living and pair bonding [1,3]. And although recent phylogenetic analyses across mammals suggest that male care is more likely a consequence of pair living than a cause [2,20], it seems to be an important factor affecting the mechanisms of pair-bond maintenance.

In sum, our study demonstrates that in red titi monkeys, females contribute more to proximity and affiliation maintenance, while males contribute more to territorial defence and infant care. Our data is most consistent with the 'male-services' hypothesis for pair-bond maintenance, where a male provides services beneficial for a female, who, in turn, shows more initiative and 'interest' in maintaining proximity and affiliation with a male. To a lesser extent, our findings also provide some support for the 'resource-defence' hypothesis, where both pair mates jointly defend their territory. Comparisons with other pair-bonded mammals suggest that male care might represent an important factor for the maintenance of pair-bonds.

Ethics. This work was conducted under all necessary permits (research permit no. 249-2017-SERFOR/DGGSPFFS from the Servicio Nacional Forestal y de Fauna Silvestre of the Peruvian Ministry of Agriculture) and ethical guidelines from the relevant authorities of Peru and the German Primate Center.

Data accessibility. The data supporting this article can be found in the electronic supplementary material.

Authors' contributions. S.D. and E.W.H. designed research. S.D. and S.W. collected data. S.D. analysed data and prepared tables and figures. S.D., E.W.H. and S.W. wrote the manuscript. All authors gave final approval for publication.

Competing interests. We declare that we have no competing interests.

Funding. This work was supported by German Primate Center, Leakey Foundation, Deutsche Forschungsgemeinschaft (DFG), International Primatological Society and Primate Action Fund.

Acknowledgements. We thank Migdonio Huanuiri Arirama, Ney Shahuano Tello, Mathieu Maréchal and all other field assistants. We are especially grateful to Camilo Flores Amasifuén without whom the fieldwork would not have been possible. We also thank the two anonymous reviewers whose suggestions helped to improve and clarify this manuscript.

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
