## [Reviewer comments · Royal Society Open Science]

Review History

RSOS-191489.R0 (Original submission)

Review form: Reviewer 1

Is the manuscript scientifically sound in its present form?

Yes

Are the interpretations and conclusions justified by the results?

No

Is the language acceptable?

Yes

Do you have any ethical concerns with this paper?

No

Have you any concerns about statistical analyses in this paper?

Yes

Recommendation?

Major revision is needed (please make suggestions in comments)

Comments to the Author(s)

This paper adds a significant amount to the literature in its detailed account of pair interactions in seven groups of wild titi monkeys.

My comments are primarily regarding the framing and interpretation of the data. A “pair bond” is a psychological construct, and the authors do a good job of distinguishing animals that may display pair bonds from those that are merely pair living. However, they then continue to make what I believe is an unnecessary distinction between pair bond maintenance and territorial defense. The pair bond itself is maintained by a number of behaviors and emotional responses, including affiliation, proximity maintenance, separation distress, and stress buffering – as well as by exclusion of potential rivals. How is male territorial defense not just another behavioral mechanism for pair bond maintenance? As such, I don’t think that the framing of “which sex invests more in the pair bond” is really the crucial question – but rather, what differing behaviors does each sex use to maintain the bond?

Another issue of framing is use of the term “pair bond strength”. In this paper, the authors are using % time in proximity as a proxy for “pair bond strength”. But – they have so much more. If they want a real measure of pair bond strength, they could use a variable reduction method to combine their multiple measures (proximity, grooming, territoriality etc.). As currently presented, I believe that the term pair bond strength is not warranted.

There are several other statements that I think need to be made a bit more carefully. For instance, pg. 6, lines 185-186, “However, females were more active in the relationships.” The method of analysis you chose suggested that males and females were equally active, so I don’t think you can make this statement. On pg. 7, lines 222-223, the sentence “females invested most heavily in grooming males during the period of infant dependency”, is also not supported by your analysis. The directionality of grooming changed after the birth of infants, but it seems like a) both pair mates reduced time grooming, and b) males reduced time grooming more than females. The statement on pg. 7 makes it sound like females increased grooming post-birth.

Another issue is that throughout, the large body of work on pair bonding and infant care in titi monkeys by Mason and Mendoza is not well-cited; many of their papers could significantly contribute to the view of titi monkey pair bonding and parenting presented here.

Review form: Reviewer 2

Is the manuscript scientifically sound in its present form?

Yes

Are the interpretations and conclusions justified by the results?

Yes

Is the language acceptable?

Yes

Do you have any ethical concerns with this paper?

No

Have you any concerns about statistical analyses in this paper?

No

Recommendation?

Accept with minor revision (please list in comments)

Comments to the Author(s)

General comments

(1) The evaluation of pair-bond strength is potentially impacted by the presence of a lactating infant. This may have implications for data presented: Brown's index data indicated females were more active in relationships. However, can the authors exclude that the greater active part of females was perhaps addressing the infant and not the male? The authors explain that males carry infants from an early age, thus is it possible that Brown's index data needs to consider and distinguish female-male dyads where males were and weren't carrying a lactating infant? It would strengthen the authors' argument if they could show quantitatively that the greater activity of females was not related to maintaining the female-offspring bond but rather the female-male bond.

(2) Data on intergroup encounters would benefit from greater clarity. First, overall sample size for encounters was surprisingly small with only 21 encounters witnessed across 7 study groups over 14 months (7 months in 2017; 7 months in 2018)! The overall encounter rate in the population was with 0.0076 enc/h (n=2750.8 observation hours across 7 groups) very low. Such low rate in itself contradicts the assumption of a significant service males could provide to females with encountering neighboring groups/individuals. The low encounter rate generally undermines the assumption and biological relevance of encounters for resource defense. Second, of observed encounters, fewer than half (42.8%) could be attributed to male initiation whereas for the majority of encounters initiations remained unknown. Given that the authors never saw a female initiate an encounter it seems unlikely that all unattributed encounters could have been initiated by females, but can it reasonably be excluded that perhaps some may have been initiated by a female? From the data presented, it seems premature to assume that males more than females initiate encounters. In this context, it remains unclear what 'initiation' really meant? Often, intergroup encounters between primate groups are characterized by agonistic interactions between neighboring individuals, which is a more straightforward measure of 'service' or 'defense' that if performed by a male would lower a female's burden since it is associated with some form of 'risk', i.e., risk of injury. Using 'initiate' rather unspecific leaves a wide spectrum of possible behaviors including behaviors of low cost, such as, for example, making the first move towards a neighboring group, and if so of equally low 'service' to a female. Moreover, in only 19% of encounters was a male seen chasing, which again seems to highlight the low-risk and thus low-service quality of male's primary engagement in intergroup encounters. Third, not all study groups were seen to engage in encounters (i.e., groups 4 & 5 are not listed in S3). Perhaps the authors can clarify if the lack of encounters in groups 4 and 5 was a result of incomplete data collection or if intergroup encounters are not observed in all titi monkey groups and thus may be condition dependent? Fourth, can the authors clarify what 'join' meant in behavioral terms, which is mentioned on P6 L193? It is such vague description that the biological relevance of the category remains unclear. Fifth, while male chasing is quantified co-chasing of a male and female is not. However, this is potentially important data needs to be reported. Co-chasing data rather supports the resource defense hypothesis as it is under this hypothesis that both male and female are expected to engage in territorial behavior. However, the authors seem to neglect data supporting the resource defense hypothesis. Sixth, females seem to have been engaged in intergroup encounters with calling. However, also this information/data is not quantified or properly reported? If calling is the main behavior females use to engage with neighboring groups than this should be given the same quality as male encounter behavior. How many encounters were witnessed in which a female was calling? Similar to my previous argument, I would

interpret female calling during intergroup encounters as a form of territorial behavior. Perhaps the authors can provide altogether more quantitative analyses of female behavior during intergroup encounters.

(3) In the first paragraph of the discussion the authors restate their observation that females were more active in pair relationship maintenance. Can they explain how 'more active' in a pair relationship translates into evolutionarily relevant behavior? The question I'm grappling with is about the biological relevance/consequence of females' greater activity? Does it mean anything in a tit-monkey relationship if one member is more active? What is the outcome of greater female activity and most importantly, do males whose females are most 'active' in the pair-bond also provide the greatest service?

Minor comments

P=page, L=line

Title: Since the correct terminology seems to be that 'males provide services', as is used in much of the manuscript, it's short form is better captured as 'males provide' instead of 'males serve', although this is a semantic suggestion.

P1L7 Change 'is rare' to 'are rare' since these are two concepts; also later i.e. 'their maintenance' instead of 'its maintenance'.

P1L32 Please, provide an example for pair living without pair bonding (following your definition P2L31) in a primate. I understand the value of differentiating pair living and pair bonding, but the problem I see lies with the definition of the pair bond as a 'long-term affiliative relationship between a male and a female'. This definition includes, for example, pair living species who live in dispersed pairs. It is hard to argue that, for example, fork-marked lemurs don't maintain 'affiliative long-term' relationships, although they forage separately and would certainly fall on the 'weak-end' of the spectrum of a pair bonded species. However, partners often stay together for a 'long-time' and when they meet, they interact affiliative. Thus, despite the dispersed pair status they still maintain 'affiliative long-term' relationships.

P2L64 You state a female can participate in territorial defense as a form of mate guarding. In addition to defense as a form of mate guarding, I suggest she can also engage in territorial defense to protect resources.

P2L67 Define 'long-term' and/or give some examples of known pair bond durations from wild titi monkeys, preferably from the studied population. Since long-term is a relative term, which moreover probably depends on a species' longevity and life history, it would be good to have some idea what time frame you have in mind when you use 'long-term'.

P2L68 Either a direct measure of 'strong' is provided or the term should be deleted. Given that a quantitative measure of pair bond strength is part of the current study it seems unnecessary to use a qualitative description here that says very little. Effectively, most if not all social relationships primates form are 'strong', because dependence and investment in social relationships is a hallmark trait of the order.

P2L78 Add article before Peruvian Amazon to read 'the Peruvian Amazon'.

P3L79 Delete 'the' before 'other'.

P3 L83 Insert an article before north-eastern Peruvian... to read 'the north-eastern Peruvian...'

P5 L163 I don't understand why rainfall is included in the list of variables with a clear impact on pair-bond strength, if shortly after this statement 'season' is shown to have no significant effect? Either some information is missing here, or it should rather read that infant presence and group size but not rainfall/season had a clear impact on pair-bond strength. Please, clarify if 'rainfall' and 'season' are intended to be synonymous?

P5 L166 Clarify what is meant with 'season'?

P6 L191 How was encounter initiation defined and systematically identified in the field?

P7 L218 It is stated that males responded 'stronger' than females to playback duets. What was the 'stronger' response? How was this measured? What is the biological relevance of a 'stronger' response? As it stands the statement is vague.

P8 L268 In my opinion, it would be more adequate and consistent with conclusions to describe male behavior as "...males provide territorial defense and infant care".

P8 L268 I think the authors neglect part of their data that supports the 'resource-defense-hypothesis' such as female participation in intergroup encounters (co-chasing, calling). Perhaps the authors can rethink their conclusion to include aspects of resource-defense in addition to male-service.

Decision letter (RSOS-191489.R0)

18-Sep-2019

Dear Ms Dolotovskaya,

The editors assigned to your paper ("Males serve, females pay: a recipe for pair bonds in a Neotropical primate?") have now received comments from reviewers. We would like you to revise your paper in accordance with the referee and Associate Editor suggestions which can be found below (not including confidential reports to the Editor). Please note this decision does not guarantee eventual acceptance.

Please submit a copy of your revised paper before 11-Oct-2019. Please note that the revision deadline will expire at 00.00am on this date. If we do not hear from you within this time then it will be assumed that the paper has been withdrawn. In exceptional circumstances, extensions may be possible if agreed with the Editorial Office in advance. We do not allow multiple rounds of revision so we urge you to make every effort to fully address all of the comments at this stage. If deemed necessary by the Editors, your manuscript will be sent back to one or more of the original reviewers for assessment. If the original reviewers are not available, we may invite new reviewers.

- Data accessibility

It is a condition of publication that all supporting data are made available either as supplementary information or preferably in a suitable permanent repository. The data

accessibility section should state where the article's supporting data can be accessed. This section should also include details, where possible of where to access other relevant research materials such as statistical tools, protocols, software etc can be accessed. If the data have been deposited in an external repository this section should list the database, accession number and link to the DOI for all data from the article that have been made publicly available. Data sets that have been deposited in an external repository and have a DOI should also be appropriately cited in the manuscript and included in the reference list.

If you wish to submit your supporting data or code to Dryad (<http://datadryad.org/>), or modify your current submission to dryad, please use the following link:
<http://datadryad.org/submit?journalID=RSOS&manu=RSOS-191489>

- **Competing interests**

- **Authors' contributions**

- **Acknowledgements**

- **Funding statement**

Kind regards,
Lianne Parkhouse
Royal Society Open Science
openscience@royalsociety.org

on behalf of Dr Alexander Ophir (Associate Editor) and Kevin Padian (Subject Editor)
openscience@royalsociety.org

Associate Editor's comments (Dr Alexander Ophir):

Dear Dr. Dolotovskaya,

I have received the reviews from two leading experts in the field of primate monogamy, and you will find their comments below. As you will see each raised some important points that I believe are crucial for you to address. For example, although Reviewer 1's comments are rather brief, they raise some very insightful points about the framing of your data and the general interpretations that follow. These are quite substantial and I believe you will need to give these careful thought as you work to address them. On the other hand, Reviewer 2's comments were more detailed, raising several points that also must be addressed, but are likely to be addressed more easily. I particularly agree with their point that additional clarity in your description of your metrics will enhance the manuscript. With this in mind, I would like to invite you to revise your manuscript to address these points satisfactorily.

All the best
Alex Ophir
Associate Editor, RSOS

Reviewers' Comments to Author:

Reviewer: 1

This paper adds a significant amount to the literature in its detailed account of pair interactions in seven groups of wild titi monkeys.

My comments are primarily regarding the framing and interpretation of the data. A "pair bond" is a psychological construct, and the authors do a good job of distinguishing animals that may display pair bonds from those that are merely pair living. However, they then continue to make what I believe is an unnecessary distinction between pair bond maintenance and territorial defense. The pair bond itself is maintained by a number of behaviors and emotional responses, including affiliation, proximity maintenance, separation distress, and stress buffering – as well as by exclusion of potential rivals. How is male territorial defense not just another behavioral mechanism for pair bond maintenance? As such, I don't think that the framing of "which sex invests more in the pair bond" is really the crucial question – but rather, what differing behaviors does each sex use to maintain the bond?

Another issue of framing is use of the term "pair bond strength". In this paper, the authors are using % time in proximity as a proxy for "pair bond strength". But – they have so much more. If they want a real measure of pair bond strength, they could use a variable reduction method to combine their multiple measures (proximity, grooming, territoriality etc.). As currently presented, I believe that the term pair bond strength is not warranted.

There are several other statements that I think need to be made a bit more carefully. For instance, pg. 6, lines 185-186, "However, females were more active in the relationships." The method of analysis you chose suggested that males and females were equally active, so I don't think you can make this statement. On pg. 7, lines 222-223, the sentence "females invested most heavily in grooming males during the period of infant dependency", is also not supported by your analysis. The directionality of grooming changed after the birth of infants, but it seems like a) both pair

mates reduced time grooming, and b) males reduced time grooming more than females. The statement on pg. 7 makes it sound like females increased grooming post-birth.

Another issue is that throughout, the large body of work on pair bonding and infant care in titi monkeys by Mason and Mendoza is not well-cited; many of their papers could significantly contribute to the view of titi monkey pair bonding and parenting presented here.

Reviewer: 2

General comments

(1) The evaluation of pair-bond strength is potentially impacted by the presence of a lactating infant. This may have implications for data presented: Brown's index data indicated females were more active in relationships. However, can the authors exclude that the greater active part of females was perhaps addressing the infant and not the male? The authors explain that males carry infants from an early age, thus is it possible that Brown's index data needs to consider and distinguish female-male dyads where males were and weren't carrying a lactating infant? It would strengthen the authors' argument if they could show quantitatively that the greater activity of females was not related to maintaining the female-offspring bond but rather the female-male bond.

(2) Data on intergroup encounters would benefit from greater clarity. First, overall sample size for encounters was surprisingly small with only 21 encounters witnessed across 7 study groups over 14 months (7 months in 2017; 7 months in 2018)! The overall encounter rate in the population was with 0.0076 enc/h (n=2750.8 observation hours across 7 groups) very low. Such low rate in itself contradicts the assumption of a significant service males could provide to females with encountering neighboring groups/individuals. The low encounter rate generally undermines the assumption and biological relevance of encounters for resource defense. Second, of observed encounters, fewer than half (42.8%) could be attributed to male initiation whereas for the majority of encounters initiations remained unknown. Given that the authors never saw a female initiate an encounter it seems unlikely that all unattributed encounters could have been initiated by females, but can it reasonably be excluded that perhaps some may have been initiated by a female? From the data presented, it seems premature to assume that males more than females initiate encounters. In this context, it remains unclear what 'initiation' really meant? Often, intergroup encounters between primate groups are characterized by agonistic interactions between neighboring individuals, which is a more straightforward measure of 'service' or 'defense' that if performed by a male would lower a female's burden since it is associated with some form of 'risk', i.e., risk of injury. Using 'initiate' rather unspecific leaves a wide spectrum of possible behaviors including behaviors of low cost, such as, for example, making the first move towards a neighboring group, and if so of equally low 'service' to a female. Moreover, in only 19% of encounters was a male seen chasing, which again seems to highlight the low-risk and thus low-service quality of male's primary engagement in intergroup encounters. Third, not all study groups were seen to engage in encounters (i.e., groups 4 & 5 are not listed in S3). Perhaps the authors can clarify if the lack of encounters in groups 4 and 5 was a result of incomplete data collection or if intergroup encounters are not observed in all titi monkey groups and thus may be condition dependent? Fourth, can the authors clarify what 'join' meant in behavioral terms, which is mentioned on P6 L193? It is such vague description that the biological relevance of the category remains unclear. Fifth, while male chasing is quantified co-chasing of a male and female is not. However, this is potentially important data needs to be reported. Co-chasing data rather supports the resource defense hypothesis as it is under this hypothesis that both male and female are expected to engage in territorial behavior. However, the authors seem to neglect data supporting the resource defense hypothesis. Sixth, females seem to have been engaged in

intergroup encounters with calling. However, also this information/data is not quantified or properly reported? If calling is the main behavior females use to engage with neighboring groups than this should be given the same quality as male encounter behavior. How many encounters were witnessed in which a female was calling? Similar to my previous argument, I would interpret female calling during intergroup encounters as a form of territorial behavior. Perhaps the authors can provide altogether more quantitative analyses of female behavior during intergroup encounters.

(3) In the first paragraph of the discussion the authors restate their observation that females were more active in pair relationship maintenance. Can they explain how 'more active' in a pair relationship translates into evolutionarily relevant behavior? The question I'm grappling with is about the biological relevance/consequence of females' greater activity? Does it mean anything in a tit-monkey relationship if one member is more active? What is the outcome of greater female activity and most importantly, do males whose females are most 'active' in the pair-bond also provide the greatest service?

Minor comments

P=page, L=line

Title: Since the correct terminology seems to be that 'males provide services', as is used in much of the manuscript, it's short form is better captured as 'males provide' instead of 'males serve', although this is a semantic suggestion.

P1L7 Change 'is rare' to 'are rare' since these are two concepts; also later i.e. 'their maintenance' instead of 'its maintenance'.

P1L32 Please, provide an example for pair living without pair bonding (following your definition P2L31) in a primate. I understand the value of differentiating pair living and pair bonding, but the problem I see lies with the definition of the pair bond as a 'long-term affiliative relationship between a male and a female'. This definition includes, for example, pair living species who live in dispersed pairs. It is hard to argue that, for example, fork-marked lemurs don't maintain 'affiliative long-term' relationships, although they forage separately and would certainly fall on the 'weak-end' of the spectrum of a pair bonded species. However, partners often stay together for a 'long-time' and when they meet, they interact affiliative. Thus, despite the dispersed pair status they still maintain 'affiliative long-term' relationships.

P2L64 You state a female can participate in territorial defense as a form of mate guarding. In addition to defense as a form of mate guarding, I suggest she can also engage in territorial defense to protect resources.

P2L67 Define 'long-term' and/or give some examples of known pair bond durations from wild titi monkeys, preferably from the studied population. Since long-term is a relative term, which moreover probably depends on a species' longevity and life history, it would be good to have some idea what time frame you have in mind when you use 'long-term'.

P2L68 Either a direct measure of 'strong' is provided or the term should be deleted. Given that a quantitative measure of pair bond strength is part of the current study it seems unnecessary to use a qualitative description here that says very little. Effectively, most if not all social relationships primates form are 'strong', because dependence and investment in social relationships is a hallmark trait of the order.

P2L78 Add article before Peruvian Amazon to read 'the Peruvian Amazon'.

P3L79 Delete 'the' before 'other'.

P3 L83 Insert an article before north-eastern Peruvian... to read 'the north-eastern Peruvian...'

P5 L163 I don't understand why rainfall is included in the list of variables with a clear impact on pair-bond strength, if shortly after this statement 'season' is shown to have no significant effect? Either some information is missing here, or it should rather read that infant presence and group size but not rainfall/season had a clear impact on pair-bond strength. Please, clarify if 'rainfall' and 'season' are intended to be synonymous?

P5 L166 Clarify what is meant with 'season'?

P6 L191 How was encounter initiation defined and systematically identified in the field?

P7 L218 It is stated that males responded 'stronger' than females to playback duets. What was the 'stronger' response? How was this measured? What is the biological relevance of a 'stronger' response? As it stands the statement is vague.

P8 L268 In my opinion, it would be more adequate and consistent with conclusions to describe male behavior as "...males provide territorial defense and infant care".

P8 L268 I think the authors neglect part of their data that supports the 'resource-defense-hypothesis' such as female participation in intergroup encounters (co-chasing, calling). Perhaps the authors can rethink their conclusion to include aspects of resource-defense in addition to male-service.

Author's Response to Decision Letter for (RSOS-191489.R0)

See Appendix A.

RSOS-191489.R1 (Revision)

Review form: Reviewer 1

Is the manuscript scientifically sound in its present form?

Yes

Are the interpretations and conclusions justified by the results?

No

Is the language acceptable?

No

Do you have any ethical concerns with this paper?

No

Have you any concerns about statistical analyses in this paper?

No

Recommendation?

Major revision is needed (please make suggestions in comments)

Comments to the Author(s)

I still find the data presented here to be interesting and a valuable contribution to the literature on a species that is still not well studied in the wild. However, I also find myself frustrated with the over-statements and lack of clarity as to what the authors are studying and what it means.

The authors obviously understand that pair bonds are psychological constructs reflected in a number of quantifiable behaviors. These concepts are based on psychological literature by

Bowlby and Ainsworth on child to parent attachment, later updated by Hazan and Shaver for adult attachment relationships. Quantifiable behaviors include a preference for the pair mate (sometimes measured by a clear choice over another potential partner; sometimes by proximity maintenance as a proxy); distress upon separation; and the ability of the partner to buffer against stressful experience. In addition, a number of related behaviors can help maintain the integrity of the bond – including mate guarding, devaluation of other potential mates (in humans), sexual behavior, and reciprocity/shared behaviors like joint child rearing. In the pair bonding literature, it is also common to talk about ANY of the above behaviors as pair bond maintenance when they occur after an initial period of formation. So while the authors stopped referring to proximity maintenance as “pair bond strength”, they are still artificially categorizing some behaviors as “pair bond maintenance” or “investment” and some as “service”. These are ALL pair bond maintenance behaviors.

I also feel like the authors are over-interpreting their data in an attempt to be flashy. I would summarize their findings as, 1) Infants reduce the time the pair mates spend in proximity. Please note that this was already found by Mendoza 1986, which is not cited here. 2) Grooming is reciprocal during periods outside of infant care. When males are carrying infants, females are slightly more responsible for grooming. 3) Proximity maintenance may be slightly female biased. 4) Chasing, although not vocal displays, during territorial encounters is male biased.

I don't know how you get from those findings to “Males serve, females pay” or “Females invest more in pair bond maintenance than males.” If anything, it is more accurate to say “Coppery titi monkeys show modest differentiation in sex-specific roles in affiliation and territorial encounters”.

Title: I didn't notice the title, actually, until Reviewer 2 pointed it out. This title should be reworded in a less provocative and flashy way.

Line 88, “titis” not “tits”

Line 121: From Mason 1966: "Rarely is an animal caught, and even when this happens the consequences are not severe. There is no extended fight; the pursuer pushes and slaps at its victim, may bite him once or twice, there are a few squeals and it is over."

While this supports the contention that fights are rarely physical, the sentence as written does not seem accurate.

Line 155: Captive coppery titi monkeys also engage in less affiliation subsequent to the birth of infants (Mendoza 1986).

Review form: Reviewer 2

Is the manuscript scientifically sound in its present form?

Yes

Are the interpretations and conclusions justified by the results?

Yes

Is the language acceptable?

Yes

Do you have any ethical concerns with this paper?

No

Have you any concerns about statistical analyses in this paper?

No

Recommendation?

Accept as is

Comments to the Author(s)

I have no further requests regarding this manuscript which will make a fine contribution to the field. My comments were addressed in full and to my satisfaction. The authors provided additional information where I had asked for it and overall increased clarity. I accept the authors' choices of comments/suggestions they rejected.

Decision letter (RSOS-191489.R1)

06-Nov-2019

Dear Ms Dolotovskaya:

Manuscript ID RSOS-191489.R1 entitled "Males serve, females pay: a recipe for pair bonds in a Neotropical primate?" which you submitted to Royal Society Open Science, has been reviewed. The comments of the reviewer(s) are included at the bottom of this letter.

Please submit a copy of your revised paper before 29-Nov-2019. Please note that the revision deadline will expire at 00.00am on this date. If we do not hear from you within this time then it will be assumed that the paper has been withdrawn. In exceptional circumstances, extensions may be possible if agreed with the Editorial Office in advance. We do not allow multiple rounds of revision so we urge you to make every effort to fully address all of the comments at this stage. If deemed necessary by the Editors, your manuscript will be sent back to one or more of the original reviewers for assessment. If the original reviewers are not available we may invite new reviewers.

Please note that it is unusual for the Editors to grant a second round of revision - if you do not satisfy them that your changes meet their requests (and those of the reviewers), it may result in the rejection of your paper in line with the journal's policy regarding revisions.

- Ethics statement

- Data accessibility

- Competing interests

- Authors' contributions

- Acknowledgements

- Funding statement

Kind regards,

Andrew Dunn

on behalf of Dr Alexander Ophir (Associate Editor) and Kevin Padian (Subject Editor)
openscience@royalsociety.org

Associate Editor Comments to Author (Dr Alexander Ophir):

Associate Editor: 1

Comments to the Author:

Dear Dr. Solotovskaya,

Your manuscript has again been seen by the two original expert referees whose reports are at the end of this email. As you can see, Reviewer 2 was satisfied by your revised draft, however Reviewer 1 continues to believe that you overstate and over interpret your results. I fully understand the enthusiasm we have for our own work, and this study is clearly exciting and important, however, I agree with Reviewer 2 that you do not need to make this paper so flashy, especially if the data do not fully substantiate these claims. I also agree that you should acknowledge the work that has demonstrated that infants reduce the time pair mates spend in proximity with each other. I find these aforementioned points compelling, and believe they will help you improve the presentation of this paper, which will require your attention during additional major revision, which I look forward to receiving.

Best

Alex Ophir
Associate Editor, RSOS

Reviewer comments to Author:

Reviewer: 1

Comments to the Author(s)

I still find the data presented here to be interesting and a valuable contribution to the literature on a species that is still not well studied in the wild. However, I also find myself frustrated with the over-statements and lack of clarity as to what the authors are studying and what it means.

The authors obviously understand that pair bonds are psychological constructs reflected in a number of quantifiable behaviors. These concepts are based on psychological literature by Bowlby and Ainsworth on child to parent attachment, later updated by Hazan and Shaver for adult attachment relationships. Quantifiable behaviors include a preference for the pair mate (sometimes measured by a clear choice over another potential partner; sometimes by proximity maintenance as a proxy); distress upon separation; and the ability of the partner to buffer against stressful experience. In addition, a number of related behaviors can help maintain the integrity of the bond – including mate guarding, devaluation of other potential mates (in humans), sexual behavior, and reciprocity/shared behaviors like joint child rearing. In the pair bonding literature, it is also common to talk about ANY of the above behaviors as pair bond maintenance when they occur after an initial period of formation. So while the authors stopped referring to proximity maintenance as “pair bond strength”, they are still artificially categorizing some behaviors as “pair bond maintenance” or “investment” and some as “service”. These are ALL pair bond maintenance behaviors.

I also feel like the authors are over-interpreting their data in an attempt to be flashy. I would summarize their findings as, 1) Infants reduce the time the pair mates spend in proximity. Please

note that this was already found by Mendoza 1986, which is not cited here. 2) Grooming is reciprocal during periods outside of infant care. When males are carrying infants, females are slightly more responsible for grooming. 3) Proximity maintenance may be slightly female biased. 4) Chasing, although not vocal displays, during territorial encounters is male biased.

I don't know how you get from those findings to "Males serve, females pay" or "Females invest more in pair bond maintenance than males." If anything, it is more accurate to say "Coppery titi monkeys show modest differentiation in sex-specific roles in affiliation and territorial encounters".

Title: I didn't notice the title, actually, until Reviewer 2 pointed it out. This title should be reworded in a less provocative and flashy way.

Line 88, "titis" not "tits"

Line 121: From Mason 1966: "Rarely is an animal caught, and even when this happens the consequences are not severe. There is no extended fight; the pursuer pushes and slaps at its victim, may bite him once or twice, there are a few squeals and it is over."

While this supports the contention that fights are rarely physical, the sentence as written does not seem accurate.

Line 155: Captive coppery titi monkeys also engage in less affiliation subsequent to the birth of infants (Mendoza 1986).

Reviewer: 2

Comments to the Author(s)

I have no further requests regarding this manuscript which will make a fine contribution to the field. My comments were addressed in full and to my satisfaction. The authors provided additional information where I had asked for it and overall increased clarity. I accept the authors' choices of comments/suggestions they rejected.

Author's Response to Decision Letter for (RSOS-191489.R1)

See Appendix B.

RSOS-191489.R2 (Revision)

Review form: Reviewer 1

Is the manuscript scientifically sound in its present form?

Yes

Are the interpretations and conclusions justified by the results?

Yes

Is the language acceptable?

Yes

Do you have any ethical concerns with this paper?

No

Have you any concerns about statistical analyses in this paper?

No

Recommendation?

Accept as is

Comments to the Author(s)

The authors have addressed my comments.

Decision letter (RSOS-191489.R2)

18-Dec-2019

Dear Ms Dolotovskaya,

It is a pleasure to accept your manuscript entitled "What makes a pair bond in a Neotropical primate: female and male contributions" in its current form for publication in Royal Society Open Science. The comments of the reviewer(s) who reviewed your manuscript are included at the foot of this letter.

Kind regards,

Andrew Dunn

on behalf of Dr Alexander Ophir (Associate Editor) and Kevin Padian (Subject Editor)
openscience@royalsociety.org

Associate Editor Comments to Author (Dr Alexander Ophir):

Associate Editor: 1

Comments to the Author:

Dear Dr. Dolotovskaya,

Thank you for your revised manuscript. I concur with the reviewer that your edits are very satisfactory and that no further edits should be necessary to proceed. Congratulations on a very nice paper.

Alex Ophir
Associate Editor, RSOS

Reviewer comments to Author:

Reviewer: 1

Comments to the Author(s)

The authors have addressed my comments.

Appendix A

Dear Dr. Ophir,

Thank you for considering our article for publication in the Royal Society Open Science. We have uploaded the revised manuscript on the journal's website. We have addressed all the comments by reviewers and made the corresponding changes to the manuscript. Detailed responses to the comments of the reviewers and the list of changes made to the manuscript are provided below.

We look forward to hearing from you regarding our submission. We would be glad to respond to any further questions and comments you may have.

Sincerely,

Sofya Dolotovskaya

Reviewer: 1

We would like to thank the reviewer for the valuable feedback. We have made the following revisions accordingly.

Comment 1: This paper adds a significant amount to the literature in its detailed account of pair interactions in seven groups of wild titi monkeys. My comments are primarily regarding the framing and interpretation of the data. A “pair bond” is a psychological construct, and the authors do a good job of distinguishing animals that may display pair bonds from those that are merely pair living. However, they then continue to make what I believe is an unnecessary distinction between pair bond maintenance and territorial defense. The pair bond itself is maintained by a number of behaviors and emotional responses, including affiliation, proximity maintenance, separation distress, and stress buffering – as well as by exclusion of potential rivals. How is male territorial defense not just another behavioral mechanism for pair bond maintenance? As such, I don't think that the framing of “which sex invests more in the pair bond” is really the crucial question – but rather, what differing behaviors does each sex use to maintain the bond?

Response: We agree with the reviewer that territorial defence can be viewed as yet another ‘pair-bond reinforcing’ behaviour and have included the paragraph addressing this framing to the revised manuscript (P2L45-52). However, as we explain in this paragraph (and also mention in the Discussion, P8), the functions of territorial behaviours can be difficult to tease apart, and not all of them might be necessarily related to the pair-bond maintenance. E.g., participation in the intergroup encounters can represent not only mate defence, but also an interest in extra-pair mates. Thus, we would prefer to address territorial behaviour separately from pair-bond maintenance behaviours — after having addressed the distinction and possible different framings in the Introduction in the revised manuscript.

Comment 2: Another issue of framing is use of the term “pair bond strength”. In this paper, the authors are using % time in proximity as a proxy for “pair bond strength”. But – they have so much more. If they want a real measure of pair bond strength, they could use a variable reduction method to combine their multiple

measures (proximity, grooming, territoriality etc.). As currently presented, I believe that the term pair bond strength is not warranted.

Response: We agree and have replaced the term 'pair-bond strength' with a simple descriptive term 'the rates of affiliation and proximity'. We see the potential of using a variable reduction method. However, we feel that the separate presentation of data on proximity, grooming etc. allows for better comparability between different studies than a composite measure where the included variables may vary between species.

Comment 3: There are several other statements that I think need to be made a bit more carefully. For instance, pg. 6, lines 185-186, "However, females were more active in the relationships." The method of analysis you chose suggested that males and females were equally active, so I don't think you can make this statement.

Response: In our opinion, this statement is justified by (1) Brown index values above 50 for all pairs, indicating that females made most of the approached and leaves; (2) the higher proportion of female approaches indicating that females made most of the approaches in all pairs. We provide both these measures in addition to Hinde index to justify this point.

Comment 4: On pg. 7, lines 222-223, the sentence "females invested most heavily in grooming males during the period of infant dependency", is also not supported by your analysis. The directionality of grooming changed after the birth of infants, but it seems like a) both pair mates reduced time grooming, and b) males reduced time grooming more than females. The statement on pg. 7 makes it sound like females increased grooming post-birth.

Response: We have now rephrased the statement as "grooming between partners was more heavily skewed towards female investment during the period of infant dependency".

Comment 5: Another issue is that throughout, the large body of work on pair bonding and infant care in titi monkeys by Mason and Mendoza is not well-cited; many of their papers could significantly contribute to the view of titi monkey pair bonding and parenting presented here.

Response: We added citations of Mason's and Mendoza's works both to the Introduction (P2) and Discussion (P7-8).

Reviewer: 2

We would like to thank the reviewer for the valuable feedback. We have made the following revisions accordingly.

General comments

Comment 1: The evaluation of pair-bond strength is potentially impacted by the presence of a lactating infant. This may have implications for data presented: Brown's index data indicated females were more active in relationships. However, can the authors exclude that the greater active part of females was perhaps

addressing the infant and not the male? The authors explain that males carry infants from an early age, thus is it possible that Brown's index data needs to consider and distinguish female-male dyads where males were and weren't carrying a lactating infant? It would strengthen the authors' argument if they could show quantitatively that the greater activity of females was not related to maintaining the female-offspring bond but rather the female-male bond.

Response: We now calculated the Brown's index separately for the periods with and without dependent infants and included this data in the Results (Table 1, P6L202-205). All the values still indicate that females made the majority of approaches and leaves in both periods.

Comment 2: Data on intergroup encounters would benefit from greater clarity. First, overall sample size for encounters was surprisingly small with only 21 encounters witnessed across 7 study groups over 14 months (7 months in 2017; 7 months in 2018)! The overall encounter rate in the population was with 0.0076 enc/h (n=2750.8 observation hours across 7 groups) very low. Such low rate in itself contradicts the assumption of a significant service males could provide to females with encountering neighboring groups/individuals. The low encounter rate generally undermines the assumption and biological relevance of encounters for resource defense.

Response: We restructured the table with the encounter account (Table S3) to improve the clarity of data. The encounter rate is indeed very low; however, it is not very different from other reports on titis: e.g., Wright (1984) only observed 7 encounters in 2 groups in 15 months. In our view, the low encounter rate does not mean that the encounters do not have any biological/evolutionary relevance for territory defence. Moreover, the encounter rate seems to be a function of the home-range overlap between the neighbouring groups: most of the encounters were observed between groups with overlapping home ranges: 2/3/6 and 1/11, while groups for which encounters were never observed (4 and 5) did not have neighbouring groups with overlapping home ranges (see also the response to Comment 5).

Comment 3: Second, of observed encounters, fewer than half (42.8%) could be attributed to male initiation whereas for the majority of encounters initiations remained unknown. Given that the authors never saw a female initiate an encounter it seems unlikely that all unattributed encounters could have been initiated by females, but can it reasonably be excluded that perhaps some may have been initiated by a female? From the data presented, it seems premature to assume that males more than females initiate encounters.

Response: Even though we cannot exclude that some of the remaining encounters could have been initiated by a female, it is very unlikely that male initiation was observed in 42% of encounters and female initiation was never observed by pure chance. Furthermore, males participated on all encounters, while females only participated in 19 encounters. Males were also more active during the encounters: in all 16 encounters for which the chasing data could be collected, males were both calling and chasing. In contrast, females mainly just called (17 encounters) and only chased during 2 encounters. We never observed a female chasing unless her mate was chasing, too. We now added this information to the Results (P6L213-219) for

better clarity and added much more detailed account of chasing and calling for both sexes to Table S3.

Comment 4: In this context, it remains unclear what 'initiation' really meant? Often, intergroup encounters between primate groups are characterized by agonistic interactions between neighboring individuals, which is a more straightforward measure of 'service' or 'defense' that if performed by a male would lower a female's burden since it is associated with some form of 'risk', i.e., risk of injury. Using 'initiate' rather unspecific leaves a wide spectrum of possible behaviors including behaviors of low cost, such as, for example, making the first move towards a neighboring group, and if so of equally low 'service' to a female. Moreover, in only 19% of encounters was a male seen chasing, which again seems to highlight the low-risk and thus low-service quality of male's primary engagement in intergroup encounters.

Response: In all titi species studied so far, intergroup encounters only consist of calling and chasing and do not involve fighting and any direct physical contact. To clarify the issue, we now added this information to the manuscript (P3L117-118) and also added the definitions of encounter initiation and participation to the Methods (P3L121-123). Thus, all the typical behaviours displayed by titis during the intergroup encounters are of lower costs than encounters that would involve physical contact and fighting which have the potential risk of injury. It does not preclude primarily male engagement in the intergroup encounters, even if this only involves relatively low-cost behaviours; more active male participation in intergroup encounters is in line with all available literature on titis (references provided in Discussion, P7L240-246).

Comment 5: Third, not all study groups were seen to engage in encounters (i.e., groups 4 & 5 are not listed in S3). Perhaps the authors can clarify if the lack of encounters in groups 4 and 5 was a result of incomplete data collection or if intergroup encounters are not observed in all titi monkey groups and thus may be condition dependent?

Response: We never observed groups 4 & 5 engaging in encounters. Most likely, it is explained by the lack of neighbouring groups with overlapping home ranges. This suggestion is indirectly supported by the notion that we had never observed intergroup encounters in group 1 until the subadult male dispersed from this group and established a new home range overlapping with the home range of group 1. We clarified it in the account of the intergroup encounters (Table S3).

Comment 6: Fourth, can the authors clarify what 'join' meant in behavioral terms, which is mentioned on P6 L193? It is such vague description that the biological relevance of the category remains unclear.

Response: We now discarded the term 'join' and used 'initiation' and 'participation' instead, with calling and chasing specified separately for better clarity and the definitions provided in Methods, P3L121-123).

Comment 7: Fifth, while male chasing is quantified co-chasing of a male and female is not. However, this is potentially important data needs to be reported. Co-chasing data rather supports the resource defense hypothesis as it is under this hypothesis

that both male and female are expected to engage in territorial behavior. However, the authors seem to neglect data supporting the resource defense hypothesis.

Response: We restructured the table with the encounter account (Table S3) (see also responses to Comments 2, 6, and 8) to clarify male and female participation. Regarding the resource defence hypothesis, we agree and discuss it now in the Discussion (P8L272-278), including the conclusions (P9L304-310).

Comment 8: Sixth, females seem to have been engaged in intergroup encounters with calling. However, also this information/data is not quantified or properly reported? If calling is the main behavior females use to engage with neighboring groups than this should be given the same quality as male encounter behavior. How many encounters were witnessed in which a female was calling? Similar to my previous argument, I would interpret female calling during intergroup encounters as a form of territorial behavior. Perhaps the authors can provide altogether more quantitative analyses of female behavior during intergroup encounters.

Response: We restructured the table with the encounter account (Table S3) and specified the participation and activity (calling/chasing) in separate columns for both sexes for each encounter.

Comment 9: In the first paragraph of the discussion the authors restate their observation that females were more active in pair relationship maintenance. Can they explain how 'more active' in a pair relationship translates into evolutionarily relevant behavior? The question I'm grappling with is about the biological relevance/consequence of females' greater activity? Does it mean anything in a tit-monkey relationship if one member is more active? What is the outcome of greater female activity and most importantly, do males whose females are most 'active' in the pair-bond also provide the greatest service?

Response: We follow Palombit (1996) interpretation of approach/leave patterns between pair mates, where the skew in the number of approaches and leaves is suggested to demonstrate an 'asymmetry of interest' in investment in the pair bond. In this context, a 'more active' (or initiative) individual is considered to be more 'interested' in pair-bond maintenance. We clarified this in the first paragraph of the discussion. In our paper, we only aimed to compare male vs. female contribution to the pair-bond maintenance, so the comparison of different pairs with different levels of female 'activity' is beyond the scope of our work.

Minor comments

Comment 1: Title: Since the correct terminology seems to be that 'males provide services', as is used in much of the manuscript, it's short form is better captured as 'males provide' instead of 'males serve', although this is a semantic suggestion.

Response: We thank the reviewer for the suggestion, but we would prefer to keep the original title as we think it would convey the main message of the paper more easily to a reader.

Comment 2: P1L7 Change 'is rare' to 'are rare' since these are two concepts; also later i.e. 'their maintenance' instead of 'its maintenance'.

Response: We fixed this.

Comment 3: P1L32 Please, provide an example for pair living without pair bonding (following your definition P2L31) in a primate. I understand the value of differentiating pair living and pair bonding, but the problem I see lies with the definition of the pair bond as a 'long-term affiliative relationship between a male and a female'. This definition includes, for example, pair living species who live in dispersed pairs. It is hard to argue that, for example, fork-marked lemurs don't maintain 'affiliative long-term' relationships, although they forage separately and would certainly fall on the 'weak-end' of the spectrum of a pair bonded species. However, partners often stay together for a 'long-time' and when they meet, they interact affiliative. Thus, despite the dispersed pair status they still maintain 'affiliative long-term' relationships.

Response: We address the difficulty of quantifying pair bonds in the next two paragraphs (P1L34-52) and provide the examples of species forming "dispersed" pairs, including the fork-marked lemurs. To further clarify these issues, we now added the definition on 'long-term' on P1L31-32.

Comment 4: P2L64 You state a female can participate in territorial defense as a form of mate guarding. In addition to defense as a form of mate guarding, I suggest she can also engage in territorial defense to protect resources.

Response: We added this notion to the revised manuscript (P2L73).

Comment 5: P2L67 Define 'long-term' and/or give some examples of known pair bond durations from wild titi monkeys, preferably from the studied population. Since long-term is a relative term, which moreover probably depends on a species' longevity and life history, it would be good to have some idea what time frame you have in mind when you use 'long-term'.

Response: We added the definition of 'long-term' on P1L31-32 and provided an example of pair-bond duration in a wild studied population of titi monkeys on P2L76-77.

Comment 6: P2L68 Either a direct measure of 'strong' is provided or the term should be deleted. Given that a quantitative measure of pair bond strength is part of the current study it seems unnecessary to use a qualitative description here that says very little. Effectively, most if not all social relationships primates form are 'strong', because dependence and investment in social relationships is a hallmark trait of the order.

Response: We agree and have deleted the term.

Comment 7: P2L78 Add article before Peruvian Amazon to read 'the Peruvian Amazon'.

Response: We fixed this.

Comment 8: P3L79 Delete 'the' before 'other'.

Response: We fixed this.

Comment 9: P3 L83 Insert an article before north-eastern Peruvian... to read 'the north-eastern Peruvian...'

Response: We fixed this.

Comment 10: P5 L163 I don't understand why rainfall is included in the list of variables with a clear impact on pair-bond strength, if shortly after this statement 'season' is shown to have no significant effect? Either some information is missing here, or it should rather read that infant presence and group size but not rainfall/season had a clear impact on pair-bond strength. Please, clarify if 'rainfall' and 'season' are intended to be synonymous?

Response: Rainfall was included in the list by mistake, we fixed this now. For clarity, we also replaced 'season' by 'rainfall' in L181, as 'rainfall' is defined in Methods section.

Comment 11: P5 L166 Clarify what is meant with 'season'?

Response: See response to the previous comment.

Comment 12: P6 L191 How was encounter initiation defined and systematically identified in the field?

Response: We added this information to the Methods (P3L121-123).

Comment 13: P7 L218 It is stated that males responded 'stronger' than females to playback duets. What was the 'stronger' response? How was this measured? What is the biological relevance of a 'stronger' response? As it stands the statement is vague.

Response: We clarified the use of 'stronger response' and rephrased the statement to better explain its relevance (P7242-245).

Comment 14: P8 L268 In my opinion, it would be more adequate and consistent with conclusions to describe male behavior as "...males provide territorial defense and infant care".

Response: We agree and changed the text accordingly.

Comment 15: P8 L268 I think the authors neglect part of their data that supports the 'resource-defense-hypothesis' such as female participation in intergroup encounters (co-chasing, calling). Perhaps the authors can rethink their conclusion to include aspects of resource-defense in addition to male-service.

Response: We agree and discuss the 'resource-defence' hypothesis in the Discussion (P8L272-278), including the conclusions (P9L307-309).

Appendix B

Dear Dr. Ophir,

Thank you very much for considering our manuscript for publication and for granting us the second round of revision. We have uploaded the revised manuscript on the journal's website. We have now changed both the framing of data and the title according to the suggestions of Reviewer 1 to avoid overstatements and overinterpretation of results. Instead of discussing pair-bond maintenance behaviours and territorial behaviours separately, we now discuss our data within the framework of differential female and male contributions to the pair-bond maintenance, as suggested by the reviewer. Unfortunately, we could not acknowledge the work demonstrating that infants reduce the time pair mates spend in proximity, as requested by the reviewer, as neither of the references provided by the reviewer contained this information (please see the detailed explanation in the reply to reviewer below, response to the last comment). Instead, we acknowledge that the result was previously found by citing another study from the field showing the same effect. Detailed responses to the reviewer's comments and the list of changes made to the manuscript are provided below.

We look forward to hearing from you regarding our submission. We would be glad to respond to any further questions and comments you may have.

Sincerely,

Sofya Dolotovskaya

Reviewer: 1

We would like to thank the reviewer for the valuable feedback. We have made the following revisions accordingly.

Comment 1: I still find the data presented here to be interesting and a valuable contribution to the literature on a species that is still not well studied in the wild. However, I also find myself frustrated with the over-statements and lack of clarity as to what the authors are studying and what it means. The authors obviously understand that pair bonds are psychological constructs reflected in a number of quantifiable behaviors. These concepts are based on psychological literature by Bowlby and Ainsworth on child to parent attachment, later updated by Hazan and Shaver for adult attachment relationships. Quantifiable behaviors include a preference for the pair mate (sometimes measured by a clear choice over another potential partner; sometimes by proximity maintenance as a proxy); distress upon separation; and the ability of the partner to buffer against stressful experience. In addition, a number of related behaviors can help maintain the integrity of the bond – including mate guarding, devaluation of other potential mates (in humans), sexual behavior, and reciprocity/shared behaviors like joint child rearing. In the pair bonding literature, it is also common to talk about ANY of the above behaviors as pair bond maintenance when they occur after an initial period of formation. So while the authors stopped referring to proximity maintenance as “pair bond strength”, they are still artificially categorizing some behaviors as “pair bond maintenance” or “investment” and some as “service”. These are ALL pair bond maintenance

behaviors. I also feel like the authors are over-interpreting their data in an attempt to be flashy. I would summarize their findings as, 1) Infants reduce the time the pair mates spend in proximity. Please note that this was already found by Mendoza 1986, which is not cited here. 2) Grooming is reciprocal during periods outside of infant care. When males are carrying infants, females are slightly more responsible for grooming. 3) Proximity maintenance may be slightly female biased. 4) Chasing, although not vocal displays, during territorial encounters is male biased. I don't know how you get from those findings to "Males serve, females pay" or "Females invest more in pair bond maintenance than males." If anything, it is more accurate to say "Coppery titi monkeys show modest differentiation in sex-specific roles in affiliation and territorial encounters".

Response: *We have changed the framing of data. Instead of discussing pair-bond maintenance behaviours and territorial behaviours separately, we now discuss our data within the framework of differential female and male contributions to pair-bond maintenance. Showing that females contribute more to the proximity and affiliation maintenance and males contribute more to territorial defence, we now discuss these findings in relation to the contributions of these behaviours to the pair bond.*

Comment 2: Title: I didn't notice the title, actually, until Reviewer 2 pointed it out. This title should be reworded in a less provocative and flashy way.

Response: *We have changed the title to "What makes a pair bond in a Neotropical primate: female and male contributions".*

Comment 3: Line 88, "titis" not "tits".

Response: *We fixed this.*

Comment 4: Line 121: From Mason 1966: "Rarely is an animal caught, and even when this happens the consequences are not severe. There is no extended fight; the pursuer pushes and slaps at its victim, may bite him once or twice, there are a few squeals and it is over." While this supports the contention that fights are rarely physical, the sentence as written does not seem accurate.

Response: *We have changed the sentence to: "...in the wild, titis very rarely engage in direct physical attacks or fighting during the encounters, even though this has been occasionally observed in captivity" to make it more accurate.*

Comment 5: Line 155: Captive coppery titi monkeys also engage in less affiliation subsequent to the birth of infants (Mendoza 1986).

Response: *Unfortunately, we could not acknowledge the work of Mendoza 1986 as requested by the reviewer. We could not find the study by Mendoza 1986, and the study by Mendoza & Mason 1986 ("Parental division of labour and differentiation of attachments in a monogamous primate (Callicebus moloch)") does not contain any information that would demonstrate that infants reduce the time the pair mates spend in proximity. To be able to acknowledge the finding, we requested the complete citation from the reviewer and were given the reference to the unpublished PhD thesis by Reeder 2001 ("The biology of parenting in monogamous titi monkey*

(Callicebus moloch)). However, after scrutinising the thesis, we still could not find any results regarding the decrease of time the pair mates spent in proximity after infant birth. The chapter addressing the relevant topic ("Study 3: Changes in inter-animal spacing across the postpartum period", pp. 74-82) only shows changes in proximity averaged over all possible combinations of animals within family groups combined, without addressing proximity in the male-female pair separately. Moreover, the effect of infant presence demonstrated in the study is opposite to the one stated by the reviewer: instead of decreasing proximity after infant birth, all family members increased proximity in the first week after infant birth, however, returning to pre-birth proximity patterns after 9 weeks: "A significant decrease in inter-animal spacing among all dyads was seen from the week prior to infant birth compared to the week after infant birth... titi monkey family members maintain closer proximity to one another following the birth of new infant and return to pre-birth spacing patterns by the time the infant is 9 weeks old" (p. 76). In sum, the study only shows the average change in proximity for all dyads combined, and these changes are only present in the first weeks after infant birth, not in the entire period of infant dependency. Therefore, the results of this study cannot be directly compared with our findings. However, to acknowledge that the result of infants reducing the time pair mates spent in proximity was found previously, we cite the field study by Spence-Aizenberg et al. 2016 that demonstrated a similar finding in wild *Plecturocebus discolor* (P7L234).